

# Inflammatory and nutritional indexes as predictors of acute kidney injury in patients with Immunoglobulin A nephropathy: a retrospective study

Huimin Li, Chuyue Qian, Jingda Huang and Mindan Sun

Department of Nephrology, The First Hospital of Jilin University, Changchun, Jilin, China

## ABSTRACT

**Background.** Immunoglobulin A nephropathy (IgAN) patients with acute kidney injury (AKI) have an elevated risk of adverse events and mortality. However, there is currently a lack of convenient and effective clinical tools to predict AKI risk in this population. The present study was conducted to create such tools containing inflammatory and nutritional indexes.

**Method.** Data from 720 adults diagnosed with IgAN by renal biopsy at the First Hospital of Jilin University were collected. They were randomly divided into a training set ($n = 503$) and a test set ($n = 217$) in a 7:3 ratio. Univariate and multivariate logistic regression analyses with backward selection were used to identify risk factors, resulting in multiple prediction models. The least absolute shrinkage and selection operator (LASSO) regression was used to simplify the model. The models were presented using nomograms, and their performances were evaluated through receiver operating characteristic (ROC) curves, area under the curve (AUC), Hosmer-Lemeshow test, net reclassification improvement (NRI), integrated discrimination improvement (IDI), calibration curves, and clinical decision curve analysis (DCA).

**Results.** Eleven risk factors related to IgAN with AKI were identified, including nephrotic syndrome (NS), T score from the Oxford histological classification, estimated glomerular filtration rate (eGFR), blood urea nitrogen (BUN), 24-hour urinary protein quantification (24h-UPRO), C-reactive protein (CRP), systemic inflammatory response index (SIRI), lymphocyte-to-monocyte ratio (LMR), platelet-to-lymphocyte ratio (PLR), lymphocyte-to-CRP ratio (LCR), and prognostic nutritional index (PNI). These factors contributed to the development of seven prediction models. ROC curves indicated good predictive performance for all models, with the full model performing best. The Hosmer-Lemeshow test showed that six models fit well in the test set. DCA results demonstrated significant clinical benefits for all models.

**Conclusion.** CRP, SIRI, LMR, PLR, LCR, and PNI were identified as novel AKI predictors in patients with IgAN. A series of prediction models incorporating these factors were developed for better clinical applicability, with the full model performing the best.

Corresponding author
Mindan Sun, sunmd@jlu.edu.cn

## INTRODUCTION

Immunoglobulin A nephropathy (IgAN) is the most common primary glomerulonephritis globally and one of the leading causes of end-stage renal disease (*Caster & Lafayette, 2024*; *Zhao et al., 2023*). It represents a clinical-pathological syndrome characterized by distinct immunopathological features but consisting of various clinical and pathological phenotypes (*Nihei et al., 2023*). The hallmark of IgAN is the deposition of immune complexes primarily composed of IgA in the mesangial area of the glomeruli, as revealed by renal biopsy pathology (*El Karoui, Fervenza & De Vriese, 2024*). Clinical manifestations can range from asymptomatic microscopic or intermittent macroscopic hematuria with stable renal function to rapidly progressive glomerulonephritis (*Stamellou et al., 2023*). Although IgAN is generally regarded as a chronic, progressive disease, some patients may experience acute kidney injury (AKI) superimposed on the underlying chronic condition (*Hogg, 2022*; *Sevillano et al., 2023*). A renal biopsy study in AKI demonstrates a high prevalence of glomerular diseases, among which IgAN is the most common (22.5%) (*Abuduwupuer et al., 2023*).

AKI is a clinical syndrome characterized by a rapid decline in kidney function and a significant increase in serum creatinine over a short period (*Jacob, Dannenhoffer & Rutter, 2020*). It is marked by diverse etiologies, complex pathologies, and severe clinical presentations (*Jacob, Dannenhoffer & Rutter, 2020*). Studies have shown that AKI is associated with an increased risk of mortality, cardiovascular events, and progression to chronic kidney disease (*Mercado, Smith & Guard, 2019*; *Schulman et al., 2023*). In patients with IgAN, the presence of AKI complicates their condition and is associated with a worse prognosis (*Oruc et al., 2017*). A previous retrospective study on IgAN found that patients with AKI exhibited more severe pathological features and significantly poorer survival outcomes compared to non-AKI patients (*Zhang, Zhuang & Liao, 2018*). Additionally, the study identified AKI as an independent risk factor for the progression of kidney disease in IgAN patients (*Zhang, Zhuang & Liao, 2018*). Identification and intervention for AKI in time can effectively prevent long-term kidney damage and significantly improve renal outcomes (*Turgut, Awad & Abdel-Rahman, 2023*). Prior research has identified potential risk factors for AKI in IgAN patients, including older age, male, malignant hypertension, proteinuria, episodes of macroscopic hematuria and pathological features (such as cellular crescents, fibrocellular crescents and glomerulosclerosis ≥50%) (*Sevillano et al., 2023*; *Sevillano et al., 2019*; *Zhang et al., 2016*). However, existing studies have yet to evaluate the relative weight of these factors, and there is a lack of convenient and effective clinical tools for the rapid identification of high-risk patients.

The neutrophil to lymphocyte ratio (NLR), lymphocyte to monocyte ratio (LMR), and platelet to lymphocyte ratio (PLR) have been shown to reflect systemic inflammatory status and immune response (*Liu et al., 2022b*) in various diseases, such as chronic kidney disease and osteoporosis (*Liu et al., 2022b*; *Wei et al., 2023*; *Zhao, Tao & Liu, 2020*). The composite inflammatory indexes systemic immune-inflammation index (SII), systemic inflammation response index (SIRI), and pan-immune inflammation value (PIV) have been demonstrated to be effective in predicting the prognosis of patients with impaired

kidney function (*Jia et al., 2022*). The ratios of monocytes, neutrophils, lymphocytes, and platelets to high-density lipoprotein cholesterol (HDL-C) can be used to comprehensively assess inflammation and lipid levels in the body (*Chen et al., 2020*; *Guo et al., 2023*; *Lu et al., 2023*). Research has found that low levels of HDL-C are an independent risk factor for AKI (*Zhou et al., 2020*). The lymphocyte to C-reactive protein ratio (LCR) is also associated with the incidence of AKI (*Song, Hu & Zhang, 2024*). Additionally, the prognostic nutritional index (PNI) and geriatric nutritional risk index (GNRI), as indicators of nutritional status, are closely related to AKI and its adverse outcomes (*Liao et al., 2023*; *Sun et al., 2024*). Thus, the present study was conducted to identify the new risk factors of AKI in IgAN patients and create clinical tools for AKI prediction. The key findings of this study are summarized in Fig. S1, which severs as a graphical abstract.

## MATERIALS & METHODS

### Study population and data source

We conducted a retrospective analysis of adults (age $\geq$18 years) diagnosed with IgAN through kidney biopsy at the First Hospital of Jilin University from January 2010 to October 2022. Among the initial 1,077 patients, 357 were excluded based on specific criteria: (1) estimated glomerular filtration rate (eGFR) <15 mL/min/1.73 m$^2$ ($n = 31$); (2) presence of autoimmune diseases ($n = 14$); (3) secondary IgAN conditions, such as hepatitis virus-related glomerulonephritis and allergic purpura nephritis ($n = 96$); (4) acute infectious diseases or cancer ($n = 14$); (5) incomplete or missing study data ($n = 202$). Ultimately, 720 eligible IgAN patients were included in the study (Fig. S2). The study protocol was thoroughly reviewed and approved by the Ethics Committee of the First Hospital of Jilin University (No. 2024-442).

### Clinical and laboratory data collection

After confirming the diagnosis of IgAN through pathology, clinical data were systematically collected including gender, age, body mass index (BMI, categorized into underweight (<18.5), normal (18.5 to 24.9), overweight (25 to 29.9), and obese ($\geq$30)); smoking status; alcohol consumption habits; history of type 2 diabetes and/or hypertension; diagnoses of nephrotic syndrome (NS); and mean arterial pressure (MAP). Laboratory data encompassed serum creatinine (SCr), eGFR (calculated using the Chronic Kidney Disease Epidemiology Collaborative (CKD-EPI) equation (*Levey et al., 2009*)), blood urea nitrogen (BUN), uric acid, serum albumin, cholinesterase, total cholesterol (TC), triglycerides (TG), HDL-C, low-density lipoprotein cholesterol (LDL-C), serum immunoglobulins (IgG, IgA, IgM), complement levels (C3, C4), Anti-streptolysin O (ASO) titer, 24-hour urinary protein quantification (24h-UPRO), urinary microalbumin (UALB), red blood cell count per high-power field (URBC), antinuclear antibody (ANA) abnormalities, C-reactive protein (CRP), and peripheral blood counts of neutrophils, monocytes, lymphocytes, and platelets. Treatment data included the use of angiotensin-converting enzyme inhibitor/angiotensin II receptor blocker (ACEI/ARB) or steroids before renal biopsy.

## Renal pathological assessment

The pathological status of each patient was evaluated using the Oxford classification system (MEST-C). This system classifies cellular/fibrocellular crescents (C), interstitial fibrosis/-tubular atrophy (T), segmental glomerulosclerosis (S), endocapillary hypercellularity (E), and mesangial hypercellularity (M) (*Trimarchi et al., 2017*).

## Clinical definition

Acute kidney injury (AKI)is defined according to the Kidney Disease (*Kellum & Lameire, 2013*): Improving Global Outcomes (KDIGO) criteria: an increase in SCr of $\geq 0.3$ mg/dl ($\geq 26.5$ $\mu$mol/l) within 48 h; an increase in SCr of $\geq 1.5$ times baseline, known or presumed to have occurred within the past 7 days; or a urine output of <0.5 ml/kg/h for more than 6 h. Due to the difficulty in obtaining urine output measurements, AKI is diagnosed based on changes in SCr levels in this study. For patients with AKI, all data were collected both before and after the onset of the condition. Each patient was hospitalized at least once between 2010 and 2020, with the first hospitalization during this period defined as the index admission. Since AKI may begin to develop prior to the index admission, the baseline SCr assessment period was extended to include outpatient values obtained before hospitalization.

Nephrotic syndrome (NS) is characterized by significant proteinuria (24-hour urinary protein excretion >3.5 g/d), hypoalbuminemia (serum albumin <30 g/L), and varying degrees of edema and hyperlipidemia (*Mizdrak et al., 2024*; *Politano, Colbert & Hamiduzzaman, 2020*). The first two criteria (proteinuria and hypoalbuminemia) are essential for diagnosis (*Mizdrak et al., 2024*; *Politano, Colbert & Hamiduzzaman, 2020*).

## Index definitions

The systemic immune-inflammation index (SII) is defined as (neutrophil count) × (platelet count)/(lymphocyte count) (*Hu et al., 2014*). The systemic inflammation response index (SIRI) is defined as (neutrophil count) × (monocyte count)/(lymphocyte count) (*Qi et al., 2016*). The pan-immune inflammation value (PIV) is defined as (neutrophil count) × (monocyte count) × (platelet count)/(lymphocyte count) (*Fucà et al., 2020*). LCR stands for lymphocyte to C-reactive protein ratio (*Minici et al., 2022*). NLR stands for neutrophil to lymphocyte ratio, LMR stands for lymphocyte to monocyte ratio, and PLR stands for platelet to lymphocyte ratio. NHR stands for neutrophil to HDL-C ratio, MHR stands for monocyte to HDL-C ratio, LHR stands for lymphocyte to HDL-C ratio, and PHR stands for platelet to HDL-C ratio.

The prognostic nutritional index (PNI) is calculated as albumin level (g/L) + 5× total lymphocyte count ($\times 10^9$/L) (*Onodera, Goseki & Kosaki, 1984*). The geriatric nutritional risk index (GNRI) is calculated as (1.489× serum albumin level (g/L)) + (41.7× weight (kg)/ideal body weight (kg)) (*Bouillanne et al., 2005*). The ideal body weight is derived using the following Lorenz equations: for males, it is calculated as height (cm) − 100 − [(height − 150)/4]; for females, it is calculated as height (cm) − 100 − [(height − 150)/2.5]. If a patient's weight exceeds the ideal body weight, the weight-to-ideal body weight ratio is set to 1 (*Bouillanne et al., 2005*).

### Composite endpoint events

Composite endpoint events were defined as a 50% decline in eGFR or eGFR <15 ml/min/1.73 m$^2$, serving as a marker for disease progression in IgAN. Follow-up time was reported in months.

### Statistical analysis

The data were analyzed using the R version 4.4.0 (http://www.R-project.org, The R Foundation). The study analysis flowchart is presented in Fig. S3. A random sampling method with a set random seed was used to divide the participants into a training set ($n = 503$) and a testing set ($n = 217$) in a 7:3 ratio. Continuous variables were described using the median (interquartile range). Categorical variables were expressed as percentages. To analyze the differences between the groups, analysis of variance (ANOVA) was used for normally distributed continuous variables, the Wilcoxon rank-sum test was used for non-normally distributed continuous variables, and the chi-square test was used for categorical variables. Univariate logistic regression analysis was first conducted to identify independent predictive variables, incorporating variables with $p < 0.05$ into the multivariate regression model. Subsequently, backward selection was employed to iteratively remove insignificant variables, stopping based on the Akaike information criterion (AIC). Multiple prediction models were then constructed based on the results. In addition, the least absolute shrinkage and selection operator (LASSO) regression was used to further simplify risk factors in the prediction model. Ten-fold cross-validation was used to select the optimal tuning parameter (lambda) which minimized the binomial deviation within one standard error of the minimum binomial deviation while also reducing model complexity. The models were constructed based on the training set and validated internally in the test set. The prediction models were presented in the form of nomograms. The predictive performance of the models was evaluated using the receiver operating characteristic curve (ROC) and the area under the curve (AUC). The net reclassification index (NRI) and integrated discrimination improvement index (IDI) were used to compare the predictive capabilities of the different models. Calibration was assessed using calibration curves and the Hosmer-Lemeshow test, $p > 0.05$ suggesting no differences between the predicted values and the actual observations. Decision curve analysis (DCA) was employed to evaluate the clinical utility of the prediction models by calculating the net benefit at various probability thresholds, determining the models' utility in real clinical settings. To enhance the reliability of the assessment, both calibration curves and DCA were performed with 1,000 iterations using the bootstrap method. Additionally, the Kaplan–Meier analysis was used to analyze the effect of the presence of AKI on the incidence of composite endpoint events in IgAN patients during the follow-up period. A significance level of 0.05 was used to determine statistical significance.

### Ethics statement

The study involving human participants was approved by the Ethics Committee of the First Hospital of Jilin University (No. 2024-442). This research was conducted in accordance with local legislation and institutional requirements. The ethics committee/institutional review

board waived the requirement for written informed consent from participants or their legal guardians/next of kin, as the data analyzed in our retrospective study was completely anonymous. Individuals whose data was being studied had no direct involvement or influence. Obtaining informed consent from each participant was deemed impractical and unnecessary, as the study posed no risks or potential harm to the subjects.

# RESULTS

## Baseline characteristics of the participants

The subjects' baseline characteristics are presented in Table 1. The median age of participants was 36.0 years, with males constituting 50% of the sample. In our study, 61 participants (8.5%) were classified as AKI. Patients with AKI had a higher proportion of males, elevated MAP, and greater prevalence of type 2 diabetes, hypertension, and NS. Laboratory results indicated that AKI patients had higher levels of SCr, BUN, uric acid, CRP, 24-hour urinary protein quantification, and urine microalbumin, along with lower levels of serum albumin, IgG, and IgM. Renal pathological showed a higher incidence of E1, T1, T2, and C2 lesions in AKI patients. Regarding immune-inflammatory indexes, AKI patients exhibited higher values of SII, SIRI, PIV, NLR, NHR, and MHR, while LCR and LMR values were lower. The PNI was lower in AKI patients, whereas the difference in GNRI between the two groups was not significant. In clinical treatment, there were no differences between the two groups in using ACEI/ARB or corticosteroids before renal biopsy. The median follow-up time of participants was 26.00 months, and 94 (13.06%) experienced composite endpoint events. Of the 659 patients without AKI, 77 (11.68%) experienced composite endpoint events, while 17 (27.87%) out of 61 patients with AKI experienced these events. IgAN patients with AKI had a significantly higher probability of experiencing composite endpoint events ($p < 0.001$).

Baseline characteristics of the training and test sets showed no statistically significant differences, as detailed in Table 2. The training set included 48 patients (9.5%) with AKI, while the test set included 13 patients (6.0%) with AKI.

## Risk factors and prediction models for AKI in patients with IgA nephropathy

Based on the univariate logistic regression analysis of the training set, 21 potential predictors were selected for multivariate logistic regression analysis (Table 3). The backward elimination method refined the predictors to 11 including NS, T, eGFR, BUN, 24h-UPRO, PNI, CRP, SIRI, LMR, PLR, and LCR. The full model was constructed with the 11 predictors, which had an AIC value of 235.6. Further, CRP, SIRI, LMR, PLR, and LCR were extracted and individually combined with other features to construct five new models as follows: model 1: NS, T, eGFR, BUN, 24h-UPRO, PNI and CRP; model 2: NS, T, eGFR, BUN, 24h-UPRO, PNI and SIRI; model 3: NS, T, eGFR, BUN, 24h-UPRO, PNI and LMR; model 4: NS, T, eGFR, BUN, 24h-UPRO, PNI and PLR; model 5: NS, T, eGFR, BUN, 24h-UPRO, PNI and LCR. In LASSO regression analysis containing the 11 risk factors, the chosen optimal tuning parameter was lambda 1se (0.03425323), resulting in the selection of eGFR, BUN, and CRP for model 6 (the process of feature screening is presented in Fig. 1).

**Table 1  Baseline characteristics of patients with IgA nephropathy.**

| Characteristic | | Overall, N = 720 | AKI | | p |
|---|---|---|---|---|---|
| | | | No, N = 659 | Yes, N = 61 | |
| Age (years) | | 36.0 (28.0, 46.0) | 36.0 (28.0, 46.0) | 40.0 (28.0, 50.0) | 0.176 |
| Gender (male), n (%) | | 361 (50.14) | 323 (49.01) | 38 (62.30) | 0.047 |
| BMI (kg/m²) | | 24.22 (21.62, 26.95) | 24.16 (21.62, 26.95) | 24.89 (21.80, 26.89) | 0.457 |
| BMI group, n (%) | | | | | 0.780 |
| Underweight (<18.5) | | 39 (5.42) | 35 (5.31) | 4 (6.56) | |
| Normal (18.5 to 24.9) | | 374 (51.94) | 345 (52.35) | 29 (47.54) | |
| Overweight (25 to 29.9) | | 251 (34.86) | 227 (34.45) | 24 (39.34) | |
| Obese (≥30) | | 56 (7.78) | 52 (7.89) | 4 (6.56) | |
| Smoking, n (%) | | 66 (9.17) | 61 (9.26) | 5 (8.20) | 0.784 |
| Drinking, n (%) | | 19 (2.64) | 16 (2.43) | 3 (4.92) | 0.213 |
| Type 2 Diabetes, n (%) | | 42 (5.83) | 34 (5.16) | 8 (13.11) | 0.020 |
| Hypertension, n (%) | | 389 (54.03) | 343 (52.05) | 46 (75.41) | <0.001 |
| NS, n (%) | | 109 (15.14) | 93 (14.11) | 16 (26.23) | 0.012 |
| MAP (mmHg) | | 103.33 (92.33, 123.33) | 102.33 (92.00, 123.00) | 120.00 (103.33, 133.33) | <0.001 |
| Renal biopsy data (Oxford MEST-C), n (%) | | | | | |
| M | 0 | 0 (0) | 0 (0) | 0 (0) | N |
| | 1 | 720 (100) | 659 (100) | 61 (100) | |
| E | 0 | 405 (56.25) | 381 (57.81) | 24 (39.34) | 0.005 |
| | 1 | 315 (43.75) | 278 (42.19) | 37 (60.66) | |
| S | 0 | 291 (40.42) | 270 (40.97) | 21 (34.43) | 0.319 |
| | 1 | 429 (59.58) | 389 (59.03) | 40 (65.57) | |
| T | 0 | 432 (60.00) | 410 (62.22) | 22 (36.07) | <0.001 |
| | 1 | 236 (32.78) | 204 (30.95) | 32 (52.46) | |
| | 2 | 52 (7.22) | 45 (6.83) | 7 (11.47) | |
| C | 0 | 404 (56.11) | 381 (57.81) | 23 (37.70) | <0.001 |
| | 1 | 233 (32.36) | 215 (32.63) | 18 (29.51) | |
| | 2 | 83 (11.53) | 63 (9.56) | 20 (32.79) | |
| Serum creatinine (μmol/L) | | 90.90 (72.40, 122.83) | 88.10 (71.00, 114.00) | 169.00 (126.10, 220.90) | <0.001 |
| eGFR (mL/min/1.73 m²) | | 80.95 (56.18, 103.13) | 83.80 (61.35, 105.95) | 39.20 (29.20, 55.80) | <0.001 |
| BUN (mmol/L) | | 6.05 (4.81, 7.69) | 5.86 (4.70, 7.36) | 8.86 (7.02, 10.78) | <0.001 |
| Serum uric acid (μmol/L) | | 394.00 (320.75, 475.00) | 390.00 (318.00, 467.50) | 443.00 (376.00, 527.00) | <0.001 |
| Serum albumin (g/L) | | 35.60 (32.00, 38.83) | 35.80 (32.10, 38.90) | 33.60 (28.70, 38.20) | 0.045 |
| Cholinesterase (U/L) | | 8,597.50 (7,329.00, 9,892.50) | 8,598.00 (7,382.50, 9,908.50) | 8,584.00 (6,435.00, 9,676.00) | 0.170 |
| TC (mmol/) | | 5.00 (4.25, 6.01) | 4.99 (4.26, 6.00) | 5.04 (4.24, 6.37) | 0.807 |
| TG (mmol/L) | | 1.62 (1.11, 2.48) | 1.60 (1.11, 2.45) | 1.84 (1.27, 2.79) | 0.055 |
| HDL-C (mmol/L) | | 1.21 (0.99, 1.47) | 1.21 (0.99, 1.48) | 1.09 (0.94, 1.39) | 0.082 |
| LDL-C (mmol/L) | | 3.12 (2.49, 3.77) | 3.12 (2.50, 3.77) | 3.06 (2.41, 3.99) | 0.672 |
| Serum IgG (g/L) | | 9.41 (7.59, 11.30) | 9.46 (7.65, 11.31) | 8.22 (6.75, 10.70) | 0.027 |
| Serum IgA (g/L) | | 2.93 (2.32, 3.74) | 2.94 (2.32, 3.74) | 2.89 (2.36, 3.53) | 0.614 |
| Serum IgM (g/L) | | 0.95 (0.69, 1.29) | 0.97 (0.69, 1.32) | 0.88 (0.52, 1.20) | 0.032 |

**Table 1** (*continued*)

| Characteristic | Overall, N = 720 | AKI | | p |
|---|---|---|---|---|
| | | No, N = 659 | Yes, N = 61 | |
| Complement 3 (g/L) | 1.10 (0.98, 1.28) | 1.10 (0.98, 1.28) | 1.07 (0.96, 1.26) | 0.271 |
| Complement 4 (g/L) | 0.27 (0.23, 0.33) | 0.27 (0.23, 0.32) | 0.28 (0.24, 0.34) | 0.136 |
| ASO (IU/ml) | 55.30 (49.63, 101.00) | 55.30 (49.70, 102.50) | 55.30 (39.91, 85.43) | 0.297 |
| 24h-UPRO (g/d) | 1.97 (1.13, 3.71) | 1.88 (1.10, 3.45) | 3.29 (1.60, 5.31) | <0.001 |
| Microalbuminuria (mg/d) | 1,428.73 (756.19, 2,917.50) | 1,368.73 (738.10, 2,814.00) | 2,263.10 (1,106.00, 4,525.00) | 0.003 |
| URBC | 14.20 (4.25, 43.23) | 14.00 (4.30, 43.00) | 17.10 (3.70, 44.50) | 0.728 |
| Abnormal ANA, n (%) | 177 (24.58) | 165 (25.04) | 12 (19.67) | 0.352 |
| CRP (mg/L) | 3.02 (1.18, 3.23) | 2.99 (1.09, 3.23) | 3.23 (2.20, 8.12) | <0.001 |
| SII | 479.52 (340.24, 655.77) | 472.43 (336.07, 649.07) | 604.18 (424.37, 781.22) | 0.003 |
| SIRI | 0.78 (0.53, 1.16) | 0.77 (0.51, 1.14) | 1.03 (0.70, 1.41) | <0.001 |
| PIV | 193.35 (120.97, 297.49) | 187.71 (118.73, 289.93) | 240.59 (165.19, 375.86) | 0.003 |
| NLR | 1.99 (1.50, 2.54) | 1.93 (1.47, 2.52) | 2.38 (1.94, 3.13) | <0.001 |
| LMR | 5.02 (3.88, 6.63) | 5.09 (3.93, 6.73) | 4.36 (3.17, 5.32) | <0.001 |
| PLR | 118.90 (97.10, 150.26) | 118.09 (96.63, 148.70) | 135.71 (107.69, 173.61) | 0.009 |
| NHR | 3.34 (2.41, 4.52) | 3.31 (2.39, 4.46) | 3.84 (2.68, 4.96) | 0.020 |
| MHR | 0.34 (0.24, 0.46) | 0.33 (0.23, 0.45) | 0.38 (0.30, 0.50) | 0.021 |
| LHR | 1.71 (1.30, 2.15) | 1.72 (1.32, 2.18) | 1.63 (1.13, 1.98) | 0.109 |
| PHR | 206.96 (155.83, 257.49) | 206.67 (155.77, 256.02) | 207.50 (165.08, 260.87) | 0.659 |
| LCR | 0.83 (0.54, 1.72) | 0.85 (0.55, 1.78) | 0.58 (0.21, 0.89) | <0.001 |
| PNI | 45.75 (41.38, 50.15) | 46.05 (41.63, 50.28) | 43.50 (36.40, 47.55) | 0.004 |
| GNRI | 93.96 (87.37, 98.78) | 93.96 (87.80, 98.88) | 91.13 (83.84, 96.50) | 0.053 |
| ACEI/ARB, n (%) | 142 (19.72) | 133 (20.18) | 9 (14.75) | 0.308 |
| Steroids, n (%) | 10 (1.39) | 9 (1.37) | 1 (1.64) | 0.590 |
| Follow-up time(months) | 26.00 (11.00, 53.25) | 27.00 (11.00, 56.50) | 21.00 (11.00, 33.00) | 0.038 |
| Composite endpoint, n (%) | 94 (13.06) | 77 (11.68) | 17 (27.87) | <0.001 |

**Notes.**

Continuous variables with non-normal distribution are presented as median (interquartile range), and categorical variables are shown as proportions n (%). The Wilcoxon rank-sum test was used to compare non-normally distributed continuous variables, while the chi-squared test was used for categorical variables.

AKI, acute kidney injury; BMI, body mass index; NS, nephrotic syndrome; MAP, mean arterial pressure; eGFR, estimated glomerular filtration rate; BUN, blood urea nitrogen; TC, total cholesterol; TG, triglyceride; HDL-C, high-density lipoprotein cholesterol; LDL-C, low-density lipoprotein cholesterol; ASO, anti-streptolysin O; 24h-UPRO, 24-hour urinary protein quantification; URBC, red blood cell count per high-power field; Abnormal ANA, abnormal antinuclear antibodies; CRP, C-reactive protein; SII, systemic immune-inflammation index; SIRI, systemic inflammation response index; PIV, pan-immune inflammation value; NLR, neutrophil to lymphocyte ratio; LMR, lymphocyte to monocyte ratio; PLR, platelet to lymphocyte ratio; NHR, neutrophil to high-density lipoprotein cholesterol ratio; MHR, monocyte to high-density lipoprotein cholesterol ratio; LHR, lymphocyte to high-density lipoprotein cholesterol ratio; PHR, platelet to high-density lipoprotein cholesterol ratio; LCR, lymphocyte to C-reactive protein ratio; PNI, prognostic nutritional index; GNRI, geriatric nutritional risk index; ACEI/ARB, angiotensin-converting enzyme inhibitor/angiotensin II receptor blocker.

Seven nomograms were developed to visually represent the interrelationships among the variables included in the prediction models (Fig. 2, Fig. S4).

## Validation and comparison of prediction models

All models demonstrated good predictive performance, as assessed by ROC curves (Fig. S5). In the training set, the AUC for the full model was 0.880 (95% CI [0.830–0.930]), while the AUC values for models 1 to 6 were 0.849 (95% CI [0.786–0.911]), 0.842 (95% CI [0.778–0.906]), 0.857 (95% CI [0.800–0.914]), 0.847 (95% CI [0.784–0.909]), 0.860 (95% CI [0.802–0.917]), and 0.841 (95% CI [0.780–0.903]), respectively. In the test set, the AUC

**Table 2  Baseline characteristics of patients in the training and testing sets.**

| Characteristic | | Train, N = 503 | Test, N = 217 | p |
|---|---|---|---|---|
| Age (years) | | 36.0 (28.0, 47.0) | 36.0 (28.0, 46.0) | 0.957 |
| Gender (male), n (%) | | 252 (50.10) | 109 (50.23) | 0.974 |
| BMI (kg/m$^2$) | | 24.22 (21.81, 26.90) | 24.17 (21.30, 27.04) | 0.370 |
| BMI group, n (%) | | | | 0.218 |
| Underweight (<18.5) | | 22 (4.38) | 17 (7.83) | |
| Normal (18.5 to 24.9) | | 266 (52.88) | 108 (49.77) | |
| Overweight (25 to 29.9) | | 173 (34.39) | 78 (35.95) | |
| Obese (≥30) | | 42 (8.35) | 14 (6.45) | |
| Smoking, n (%) | | 42 (8.35) | 24 (11.06) | 0.248 |
| Drinking, n (%) | | 12 (2.39) | 7 (3.23) | 0.519 |
| Type 2 Diabetes, n (%) | | 32 (6.36) | 10 (4.61) | 0.357 |
| Hypertension, n (%) | | 279 (55.47) | 110 (50.69) | 0.238 |
| NS, n (%) | | 73 (14.51) | 36 (16.59) | 0.476 |
| MAP (mmHg) | | 103.33 (92.50, 123.33) | 102.00 (91.00, 124.67) | 0.654 |
| Renal biopsy data (Oxford MEST-C), n (%) | | | | |
| M | 0 | 0 (0) | 0 (0) | N |
| | 1 | 503 (100) | 217 (100) | |
| E | 0 | 289 (57.46) | 116 (53.46) | 0.321 |
| | 1 | 214 (42.54) | 101 (46.54) | |
| S | 0 | 205 (40.76) | 86 (39.63) | 0.778 |
| | 1 | 298 (59.24) | 131 (60.37) | |
| T | 0 | 307 (61.03) | 125 (57.60) | 0.364 |
| | 1 | 164 (32.60) | 72 (33.18) | |
| | 2 | 32 (6.36) | 20 (9.22) | |
| C | 0 | 280 (55.67) | 124 (57.14) | 0.744 |
| | 1 | 162 (32.21) | 71 (32.72) | |
| | 2 | 61 (12.12) | 22 (10.14) | |
| Serum creatinine (μmol/L) | | 90.80 (71.75, 122.95) | 91.00 (72.80, 122.20) | 0.805 |
| eGFR (mL/min/1.73 m$^2$) | | 80.40 (56.00, 103.20) | 81.70 (58.30, 103.00) | 0.716 |
| BUN (mmol/L) | | 6.10 (4.77, 7.69) | 6.00 (4.91, 7.70) | 0.808 |
| Serum uric acid (μmol/L) | | 395.00 (319.00, 474.50) | 389.00 (324.00, 475.00) | 0.997 |
| Serum albumin (g/L) | | 35.60 (32.10, 38.60) | 35.80 (31.60, 39.30) | 0.816 |
| Cholinesterase (U/L) | | 8,584.00 (7,356.50, 9,953.00) | 8,685.00 (7,267.00, 9,842.00) | 0.781 |
| TC (mmol/) | | 4.99 (4.20, 6.05) | 5.03 (4.35, 5.85) | 0.615 |
| TG (mmol/L) | | 1.62 (1.12, 2.49) | 1.56 (1.11, 2.45) | 0.645 |
| HDL-C (mmol/L) | | 1.19 (0.98, 1.46) | 1.25 (1.01, 1.50) | 0.117 |
| LDL-C (mmol/L) | | 3.09 (2.49, 3.80) | 3.13 (2.51, 3.68) | 0.955 |
| Serum IgG (g/L) | | 9.27 (7.60, 11.30) | 9.62 (7.54, 11.23) | 0.750 |
| Serum IgA (g/L) | | 2.93 (2.32, 3.73) | 2.94 (2.31, 3.74) | 0.881 |
| Serum IgM (g/L) | | 0.97 (0.69, 1.33) | 0.93 (0.66, 1.22) | 0.374 |
| Complement 3 (g/L) | | 1.09 (0.97, 1.28) | 1.11 (0.98, 1.29) | 0.475 |

| Characteristic | Train, N = 503 | Test, N = 217 | p |
|---|---|---|---|
| Complement 4 (g/L) | 0.27 (0.23, 0.33) | 0.28 (0.23, 0.33) | 0.863 |
| ASO (IU/ml) | 55.00 (49.40, 98.40) | 55.30 (49.70, 106.27) | 0.259 |
| 24h-UPRO (g/d) | 1.94 (1.12, 3.45) | 2.04 (1.17, 3.99) | 0.391 |
| Microalbuminuria (mg/d) | 1,428.70 (734.00, 2,827.00) | 1,460.00 (817.60, 3,199.50) | 0.535 |
| URBC | 13.50 (4.00, 44.90) | 16.10 (5.20, 40.30) | 0.686 |
| Abnormal ANA, n (%) | 124 (24.65) | 53 (24.42) | 0.948 |
| CRP (mg/L) | 3.02 (1.22, 3.23) | 3.02 (1.07, 3.23) | 0.844 |
| SII | 478.29 (338.24, 649.19) | 481.38 (343.94, 684.13) | 0.724 |
| SIRI | 0.78 (0.52, 1.21) | 0.78 (0.54, 1.11) | 0.588 |
| PIV | 194.73 (123.14, 301.00) | 191.98 (118.55, 288.83) | 0.773 |
| NLR | 1.97 (1.50, 2.54) | 1.99 (1.52, 2.54) | 0.888 |
| LMR | 4.94 (3.88, 6.57) | 5.14 (3.83, 6.70) | 0.420 |
| PLR | 118.43 (96.80, 148.54) | 121.23 (98.05, 153.55) | 0.451 |
| NHR | 3.39 (2.44, 4.59) | 3.25 (2.32, 4.43) | 0.369 |
| MHR | 0.35 (0.24, 0.47) | 0.32 (0.23, 0.44) | 0.089 |
| LHR | 1.73 (1.33, 2.21) | 1.69 (1.27, 2.05) | 0.260 |
| PHR | 208.87 (158.07, 257.65) | 202.92 (154.08, 252.17) | 0.402 |
| LCR | 0.85 (0.54, 1.73) | 0.81 (0.52, 1.67) | 0.598 |
| PNI | 45.40 (41.53, 50.15) | 46.15 (41.00, 50.30) | 0.978 |
| GNRI | 93.67 (87.81, 98.28) | 94.18 (85.81, 99.20) | 0.982 |
| ACEI/ARB, n (%) | 91 (18.09) | 51 (23.50) | 0.094 |
| Steroids, n (%) | 6 (1.19) | 4 (1.84) | 0.499 |
| AKI, n (%) | 48 (9.54) | 13 (5.99) | 0.116 |

**Notes.**

Continuous variables with non-normal distribution are presented as median (interquartile range), and categorical variables are shown as proportions n (%). The Wilcoxon rank-sum test was used to compare non-normally distributed continuous variables, while the chi-squared test was used for categorical variables.

AKI, acute kidney injury; BMI, body mass index; NS, nephrotic syndrome; MAP, mean arterial pressure; eGFR, estimated glomerular filtration rate; BUN, blood urea nitrogen; TC, total cholesterol; TG, triglyceride; HDL-C, high-density lipoprotein cholesterol; LDL-C, low-density lipoprotein cholesterol; ASO, anti-streptolysin O; 24h-UPRO, 24-hour urinary protein quantification; URBC, red blood cell count per high-power field; Abnormal ANA, abnormal antinuclear antibodies; CRP, C-reactive protein; SII, systemic immune-inflammation index; SIRI, systemic inflammation response index; PIV, pan-immune inflammation value; NLR, neutrophil to lymphocyte ratio; LMR, lymphocyte to monocyte ratio; PLR, platelet to lymphocyte ratio; NHR, neutrophil to high-density lipoprotein cholesterol ratio; MHR, monocyte to high-density lipoprotein cholesterol ratio; LHR, lymphocyte to high-density lipoprotein cholesterol ratio; PHR, platelet to high-density lipoprotein cholesterol ratio; LCR, lymphocyte to C-reactive protein ratio; PNI, prognostic nutritional index; GNRI, geriatric nutritional risk index; ACEI/ARB, angiotensin-converting enzyme inhibitor/angiotensin II receptor blocker.

for the full model was 0.846 (95% CI [0.748–0.944]), while the AUC values for models 1 to 6 were 0.884 (95% CI [0.829–0.938]), 0.875 (95% CI [0.819–0.930]), 0.867 (95% CI [0.809–0.924]), 0.885 (95% CI [0.837–0.934]), 0.848 (95% CI [0.732–0.964]), and 0.889 (95% CI [0.833–0.945]), respectively. Compared to the full model, the AUC values for the other models were lower in the training set but higher in the test set.

NRI and IDI of the other models relative to the full model are presented in Table 4. When using a cutoff value of 10% for AKI occurrence probability, models 1 to 6 did not improve the predictive capability compared to the full model. The IDI values for Models 1 to 6 were less than 0, showing no significant enhancement of overall diagnostic capability compared to the full model.

The calibration curves for all models are shown in Fig. 3. In the training set, the Hosmer-Lemeshow test confirmed adequate goodness of fit for the full model and model

**Table 3  Univariate logistic regression analysis of the training set data.**

| Characteristic | | Univariable analysis OR (95%CI) | p |
|---|---|---|---|
| Age (years) | | 1.03 (1.00–1.05) | 0.027 |
| Gender (male) | | 0.52 (0.28–0.96) | 0.037 |
| BMI (kg/m$^2$) | | 1.00 (0.93–1.08) | 0.954 |
| BMI group | | | |
| Underweight (<18.5) | | Ref | |
| Normal (18.5 to 24.9) | | 0.63 (0.17–2.28) | 0.479 |
| Overweight (25 to 29.9) | | 0.69 (0.19–2.57) | 0.581 |
| Obese ($\geq$30) | | 0.67 (0.14–3.29) | 0.618 |
| Smoking | | 0.71 (0.21–2.39) | 0.582 |
| Drinking | | 1.93 (0.41–9.10) | 0.403 |
| Type 2 Diabetes | | 2.36 (0.92–6.05) | 0.075 |
| Hypertension | | 2.33 (1.20–4.52) | 0.012 |
| NS | | 2.45 (1.22–4.89) | 0.011 |
| MAP (mmHg) | | 1.02 (1.01–1.03) | 0.002 |
| Renal biopsy data (Oxford MEST-C) | | | |
| E | 0 | Ref | |
| | 1 | 1.68 (0.92–3.05) | 0.089 |
| S | 0 | Ref | |
| | 1 | 1.42 (0.76–2.66) | 0.273 |
| T | 0 | Ref | |
| | 1 | 2.22 (1.19–4.15) | 0.012 |
| | 2 | 1.95 (0.62–6.07) | 0.252 |
| C | 0 | Ref | |
| | 1 | 1.30 (0.63–2.67) | 0.475 |
| | 2 | 4.48 (2.12–9.44) | <0.001 |
| Serum creatinine ($\mu$mol/L) | | 1.02 (1.01–1.03) | <0.001 |
| eGFR (mL/min/1.73 m$^2$) | | 0.95 (0.94–0.96) | <0.001 |
| BUN (mmol/L) | | 1.36 (1.23–1.50) | <0.001 |
| Serum uric acid ($\mu$mol/L) | | 1.00 (1.00–1.01) | 0.002 |
| Serum albumin (g/L) | | 0.97 (0.93–1.02) | 0.223 |
| Cholinesterase (U/L) | | 1.00 (1.00–1.00) | 0.644 |
| TC (mmol/) | | 1.10 (0.95–1.27) | 0.208 |
| TG (mmol/L) | | 1.12 (0.95–1.31) | 0.166 |
| HDL-C (mmol/L) | | 0.56 (0.24–1.32) | 0.184 |
| LDL-C (mmol/L) | | 1.10 (0.89–1.36) | 0.384 |
| Serum IgG (g/L) | | 0.94 (0.85–1.04) | 0.226 |
| Serum IgA (g/L) | | 1.12 (0.90–1.40) | 0.321 |
| Serum IgM (g/L) | | 0.41 (0.20–0.85) | 0.017 |
| Complement 3 (g/L) | | 0.49 (0.14–1.76) | 0.276 |
| Complement 4 (g/L) | | 4.47 (0.15–135.65) | 0.390 |

**Table 3** (*continued*)

| Characteristic | Univariable analysis OR (95%CI) | *p* |
|---|---|---|
| ASO (IU/ml) | 1.00 (0.99–1.0) | 0.214 |
| 24h-UPRO (g/d) | 1.10 (1.02–1.18) | 0.016 |
| Microalbuminuria (mg/d) | 1.00 (1.00–1.00) | 0.065 |
| URBC | 1.00 (1.00–1.00) | 0.743 |
| Abnormal ANA | 0.68 (0.32–1.45) | 0.321 |
| CRP (mg/L) | 1.06 (1.02–1.09) | 0.001 |
| SII | 1.00 (1.00–1.00) | 0.143 |
| SIRI | 1.59 (1.08–2.34) | 0.018 |
| PIV | 1.00 (1.00–1.00) | 0.094 |
| NLR | 1.43 (1.10–1.87) | 0.009 |
| LMR | 0.70 (0.58–0.85) | <0.001 |
| PLR | 1.01 (1.00–1.01) | 0.033 |
| NHR | 1.17 (0.99–1.37) | 0.061 |
| MHR | 6.11 (1.51–24.62) | 0.011 |
| LHR | 0.87 (0.58–1.32) | 0.520 |
| PHR | 1.00 (1.00–1.01) | 0.447 |
| LCR | 0.39 (0.22–0.68) | 0.001 |
| PNI | 0.96 (0.93–1.00) | 0.045 |
| GNRI | 0.98 (0.95–1.01) | 0.183 |
| ACEI/ARB | 0.50 (0.19–1.30) | 0.154 |
| Steroids | 1.91 (0.22–16.74) | 0.557 |

**Notes.**

BMI, body mass index; NS, nephrotic syndrome; MAP, mean arterial pressure; eGFR, estimated glomerular filtration rate; BUN, blood urea nitrogen; TC, total cholesterol; TG, triglyceride; HDL-C, high-density lipoprotein cholesterol; LDL-C, low-density lipoprotein cholesterol; ASO, anti-streptolysin O; 24h-UPRO, 24-hour urinary protein quantification; URBC, red blood cell count per high-power field; Abnormal ANA, abnormal antinuclear antibodies; CRP, C-reactive protein; SII, systemic immune-inflammation index; SIRI, systemic inflammation response index; PIV, pan-immune inflammation value; NLR, neutrophil to lymphocyte ratio; LMR, lymphocyte to monocyte ratio; PLR, platelet to lymphocyte ratio; NHR, neutrophil to high-density lipoprotein cholesterol ratio; MHR, monocyte to high-density lipoprotein cholesterol ratio; LHR, lymphocyte to high-density lipoprotein cholesterol ratio; PHR, platelet to high-density lipoprotein cholesterol ratio; LCR, lymphocyte to C-reactive protein ratio; PNI, prognostic nutritional index; GNRI, geriatric nutritional risk index; ACEI/ARB, angiotensin-converting enzyme inhibitor/angiotensin II receptor blocker.

1 to 6 (*p*-values: 0.764, 0.326, 0.518, 0.087, 0.161, 0.467, 0.131, respectively). In the test set, the full model, model 1 to 4, and model 6 demonstrated good calibration (*p*-values: 0.261, 0.119, 0.365, 0.296, 527, 0.499, respectively), while the model 5 exhibited poor calibration (*p* = 0.002, Table S1).

Finally, the DCA indicated that within a certain range of threshold probabilities, all models demonstrated a net benefit greater than 0, with model curves yielding higher benefits compared to the two extreme curves. Notably, the complete model and model 1 exhibited a wide range of threshold probabilities. This suggests that the prediction models mentioned above may have practical application value in various clinical scenarios (Fig. 4).

To assess the independent predictive value of the composite indexes, we compared their performance with that of their individual components using ROC curve analysis. The results indicated that the composite indexes (PNI, SIRI, LMR, PLR, LCR) generally demonstrated superior predictive performance compared to their components (Table S2).

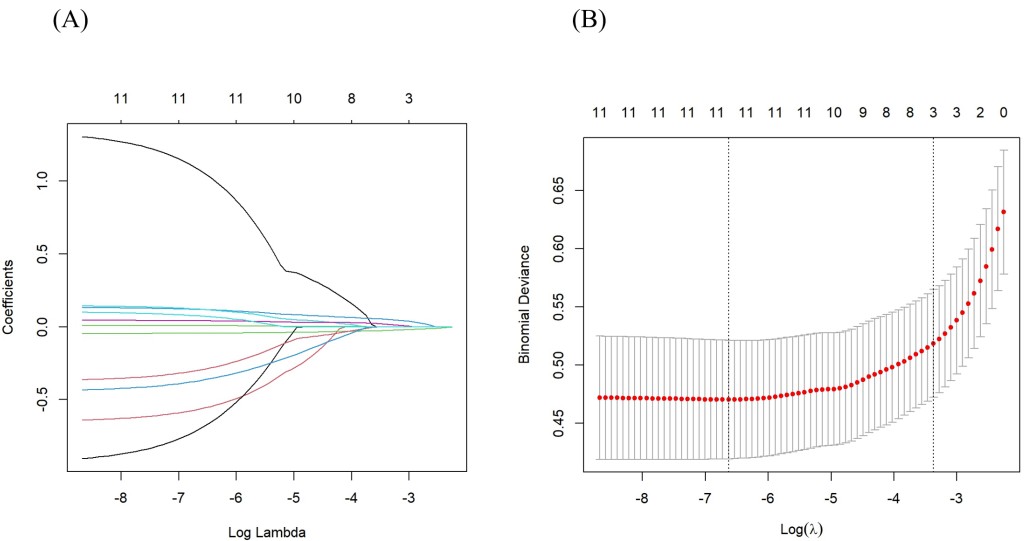

(A)

(B)

**Figure 1 Features were screened using LASSO regression analysis.** (A) Distribution of LASSO coefficients for the 11 risk factors. (B) Relationship between binomial deviance curve and log ($\lambda$), where $\lambda$ denotes the lambda value. In (B), The two vertical dashed lines in the figure are drawn based on the optimal scores for the minimum lambda and lambda.1se standards. The minimum lambda minimizes the binomial deviance, while lambda.1se keeps the binomial deviance within one standard error of the minimum binomial deviance. LASSO, least absolute shrinkage and selection operator.

These results emphasize the superior predictive value of the composite indexes, supporting their inclusion in AKI risk prediction models for IgAN patients.

## The association between AKI and long-term prognosis in IgAN patients

Kaplan–Meier analysis showed that IgAN patients without AKI had a higher progression-free survival probability compared to those with AKI (Log-rank $p < 0.01$; Fig. S6).

## DISCUSSION

### Main findings

IgAN patients who develop AKI face an increased risk of adverse events and mortality (*Zhang, Zhuang & Liao, 2018*). Consistent with previous studies, our findings indicated that AKI is closely associated with poor long-term prognosis in IgAN patients. Therefore, it is crucial to focus on the early identification of IgAN patients at risk of AKI and to implement preventive measures, rather than solely concentrating on treating those with established AKI. This study identified 11 risk factors for the occurrence of AKI in IgAN patients, which included the presence of NS, T1 in the Oxford histological score, high levels of BUN, 24h-UPRO, CRP, SIRI, and PLR, as well as low levels of eGFR, LMR, LCR, and PNI. Based on these findings, we constructed a comprehensive model for predicting the risk of AKI in IgAN patients. Since CRP, SIRI, LMR, PLR, and LCR are primarily related to inflammation, these inflammatory markers were extracted and individually combined with other features to construct five new prediction models, extending the

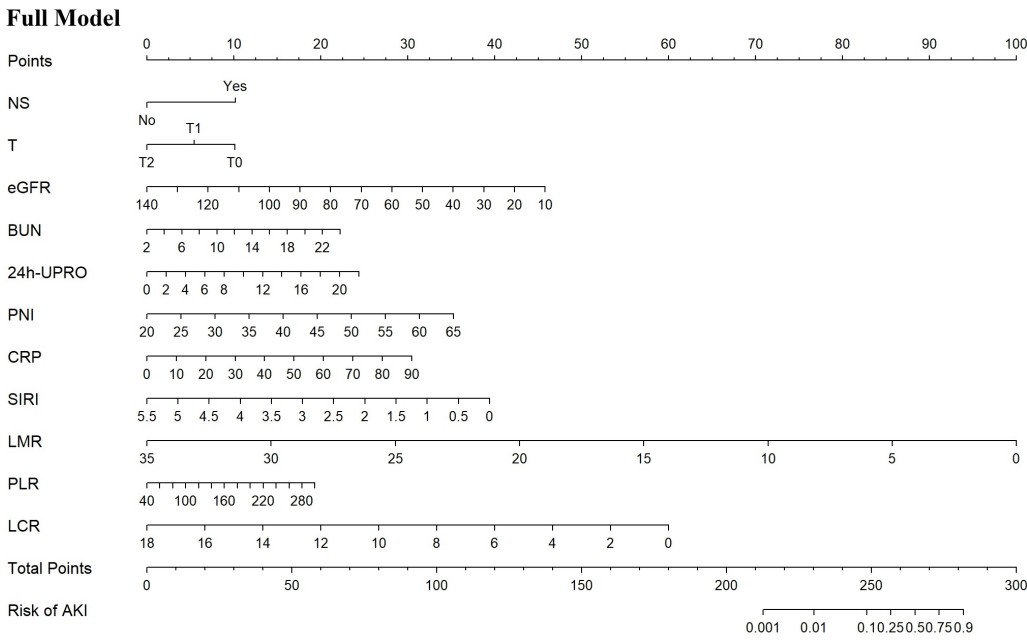

**Figure 2  Nomograms predicting the probability of AKI in IgAN patients for the full model.** AKI, acute kidney injury; IgAN, Immunoglobulin A nephropathy; NS, nephrotic syndrome; T, interstitial fibrosis/tubular atrophy; eGFR, estimated glomerular filtration rate; BUN, blood urea nitrogen; 24h-UPRO, 24-hour urinary protein quantification; PNI, prognostic nutritional index; CRP, C-reactive protein; SIRI, systemic inflammation response index; LMR, lymphocyte to monocyte ratio; PLR, platelet to lymphocyte ratio; LCR, lymphocyte to C-reactive protein ratio.

usability of the models. Additionally, we employed LASSO regression to select the risk factors, identifying eGFR, BUN, and CRP, thereby developing a more streamlined model (model 6). Consequently, seven prediction models were developed, each accompanied by a nomogram to quantify the risk of AKI. All models demonstrated good predictive capabilities, with the full model being the best performer. All models, except model 5, exhibited good calibration. We also discovered that all models possessed strong clinical decision-making abilities. In summary, for clinical prediction of AKI risk in IgAN patients, we recommend adopting the full model as the first-line tool. Other models, excluding model 5, remain viable alternatives based on specific clinical needs.

## Analysis of risk factors in the prediction models

Our prediction models incorporate various factors, including nephrotic syndrome (NS), proteinuria, glomerular filtration rate, nutrition, inflammation, and renal pathological changes. This study found that NS in IgAN patients is one of the risk factors for AKI. Existing evidence reports a 39% AKI incidence among adult NS patients during the initial four-month period (*Kolb et al., 2021*). Current evidence indicates that the occurrence of AKI in NS patients may involve several mechanisms, including renal ischemia, tubular necrosis, interstitial edema, renal vein thrombosis, drug-related interstitial nephritis, and crescentic glomerulonephritis (*Lionaki, Liapis & Boletis, 2019*; *Lu et al., 2022*). NS is a relatively rare but serious manifestation of IgAN (*Chen et al., 2022*). In IgAN patients, those with

**Table 4  The NRI and IDI of models 1 to 6 compared to the complete model in predicting AKI risk in patients with IgA nephropathy.**

| | Training set | | Test set | |
|---|---|---|---|---|
| | Index (95% CI) | *p* | Index (95% CI) | *p* |
| **Model 1** | | | | |
| NRI | 0.022 (−0.064, 0.108) | 0.619 | 0.077 (−0.185, 0.339) | 0.565 |
| IDI | −0.049 (−0.083, −0.016) | 0.004 | −0.012 (−0.078, 0.054) | 0.722 |
| **Model 2** | | | | |
| NRI | 0.015 (−0.089, 0.119) | 0.773 | −0.033 (−0.195, 0.129) | 0.689 |
| IDI | −0.076 (−0.124, −0.027) | 0.002 | −0.029 (−0.083, 0.025) | 0.293 |
| **Model 3** | | | | |
| NRI | 0.035 (−0.051, 0.121) | 0.422 | −0.033 (−0.184, 0.118) | 0.665 |
| IDI | −0.062 (−0.106, −0.017) | 0.007 | −0.021 (−0.071, 0.030) | 0.423 |
| **Model 4** | | | | |
| NRI | −0.012 (−0.108, 0.084) | 0.805 | 0.00 (−0.161, 0.161) | 1.000 |
| IDI | −0.059 (−0.102, −0.016) | 0.007 | −0.015 (−0.059, 0.029) | 0.514 |
| **Model 5** | | | | |
| NRI | 0.016 (−0.059, 0.092) | 0.669 | −0.033 (−0.182, 0.115) | 0.660 |
| IDI | −0.043 (−0.080, −0.006) | 0.023 | −0.022 (−0.057, 0.013) | 0.219 |
| **Model 6** | | | | |
| NRI | −0.019 (−0.097, 0.059) | 0.637 | −0.200 (−0.325, −0.075) | 0.002 |
| IDI | −0.097 (−0.144, −0.049) | <0.001 | −0.112 (−0.159, −0.064) | <0.001 |

Notes.
Model 1, NS, T, eGFR, BUN, 24h-UPRO, PNI and CRP.
Model 2, NS, T, eGFR, BUN, 24h-UPRO, PNI and SIRI.
Model 3, NS, T, eGFR, BUN, 24h-UPRO, PNI and LMR.
Model 4, NS, T, eGFR, BUN, 24h-UPRO, PNI and PLR.
Model 5, NS, T, eGFR, BUN, 24h-UPRO, PNI and LCR;
Model 6, eGFR, BUN and CRP. NS, nephrotic syndrome; T, interstitial fibrosis/tubular atrophy.
eGFR, estimated glomerular filtration rate; BUN, blood urea nitrogen; 24h-UPRO, 24-hour urinary protein quantification; PNI, prognostic nutritional index; CRP, C-reactive protein; SIRI, systemic inflammation response index; LMR, lymphocyte to monocyte ratio; PLR, platelet to lymphocyte ratio; LCR, lymphocyte to C-reactive protein ratio; NRI, net reclassification improvement index; IDI, integrated discrimination improvement index.

comorbid NS exhibit more diffuse foot process effacement in electron microscopy (*Jiang et al., 2024*). Podocyte injury and loss are critical factors contributing to the progressive of proteinuria and renal filtration dysfunction in IgAN (*Chen et al., 2023*).

An observational study found that the risk of hospitalization due to AKI increased with higher levels of proteinuria and lower eGFR (*James et al., 2010*). Consistent with previous studies, our research identified 24-hour urinary protein quantification and eGFR as predictive factors for AKI in IgAN patients. From a pathophysiological perspective, proteinuria reflects dysfunction in the size-selective properties of the glomerular barrier or may arise from tubular-interstitial pathology (*Miceli, 2015*). Urinary protein reabsorption by the renal tubules triggers pro-inflammatory signaling, leading to tubulointerstitial injury (*Jiang et al., 2023*). This damage undermines the kidney's ability to tolerate hemodynamic fluctuations and nephrotoxins insults, contributing to the onset of AKI (*Jiang et al., 2023*). Another essential feature of NS, hypoalbuminemia, can lead to third-space fluid retention

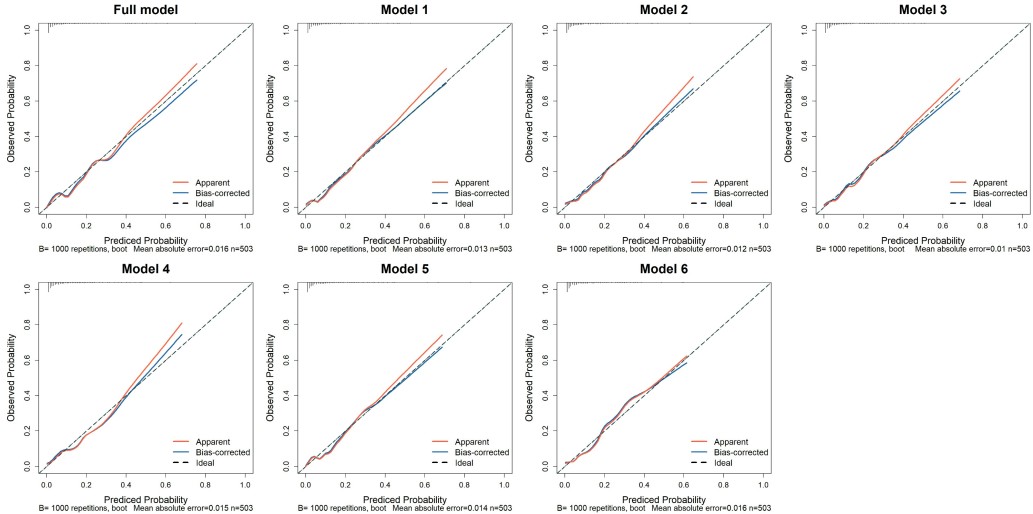

**Figure 3** **The calibration curves of the seven prediction models in the training set.** The *x*-axis represents the predicted probability of AKI, while the *y*-axis indicates the actual probability of AKI. The red solid line, "Apparent," reflects the consistency between the calculated risk probabilities based on the model and the actual probabilities. The blue solid line, "Bias-corrected," represents the results obtained after bootstrapping the model data 1,000 times. The diagonal dashed line illustrates the perfect prediction of an ideal model. Full model: NS, T, eGFR, BUN, 24h-UPRO, PNI, CRP, SIRI, LMR, PLR, and LCR; Model 1: NS, T, eGFR, BUN, 24h-UPRO, PNI and CRP; Model 2: NS, T, eGFR, BUN, 24h-UPRO, PNI and SIRI; Model 3: NS, T, eGFR, BUN, 24h-UPRO, PNI and LMR; Model 4: NS, T, eGFR, BUN, 24h-UPRO, PNI and PLR; Model 5: NS, T, eGFR, BUN, 24h-UPRO, PNI and LCR; Model 6: eGFR, BUN and CRP. AKI, acute kidney injury; NS, nephrotic syndrome; T, interstitial fibrosis/tubular atrophy; eGFR, estimated glomerular filtration rate; BUN, blood urea nitrogen; 24h-UPRO, 24-hour urinary protein quantification; PNI, prognostic nutritional index; CRP, C-reactive protein; SIRI, systemic inflammation response index; LMR, lymphocyte to monocyte ratio; PLR, platelet to lymphocyte ratio; LCR, lymphocyte to C-reactive protein ratio.

and a reduction in effective circulating blood volume, thereby resulting in AKI (*Lin et al., 2020*). Decreased serum albumin levels may also increase blood viscosity and impair endothelial function (*Kurtul, Gok & Esenboga, 2021*).

Minimal change disease patients with AKI demonstrated significantly lower serum albumin levels compared with those without AKI (*Lin et al., 2020*). Albumin is an important indicator for assessing nutritional status (*Dong et al., 2021*). To further investigate the relationship between nutrition and AKI, we incorporated PNI and GNRI. These indexes were used to assess the patient's nutritional status and inflammatory state (*Liu et al., 2022a*; *Onodera, Goseki & Kosaki, 1984*; *Ruan et al., 2023*). However, among the two indices, only PNI was demonstrated to be a risk factor for AKI in IgAN patients. Multiple studies have indicated that a low PNI score is a risk factor for AKI (*Chen et al., 2024c*; *Liu et al., 2023*; *Zhu et al., 2023*). PNI has also been validated as an independent predictor of AKI development in critically ill cohorts (*Chen et al., 2024b*). Decreased albumin synthesis, increased catabolism, and exacerbated inflammation can lead to malnutrition (*Dong et al., 2021*). Malnutrition can cause cellular and tissue damage, promoting oxidative stress and inflammatory processes, ultimately leading to kidney injury (*Sun et al., 2024*).

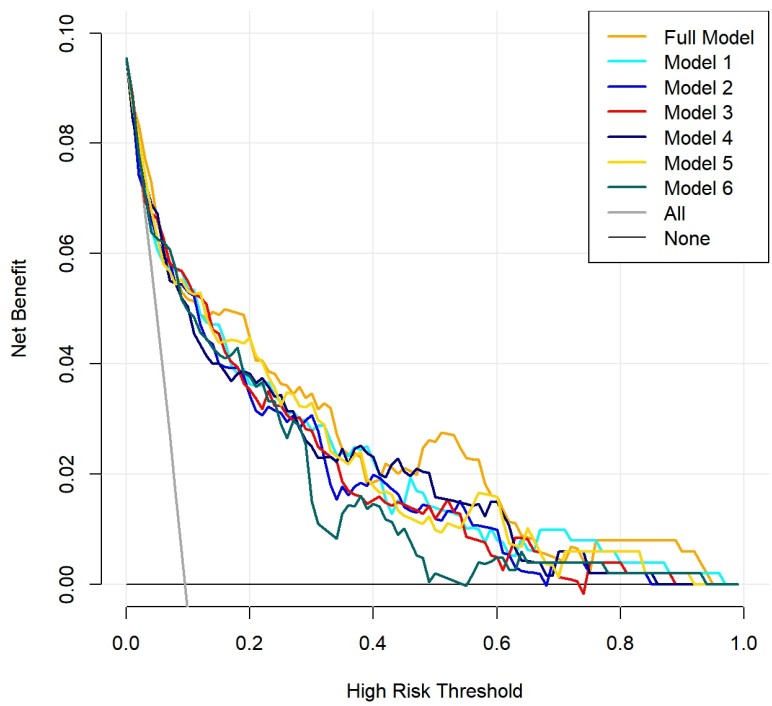

**Figure 4 Clinical decision curve analysis (DCA) of the seven prediction models.** The red solid line represents the net benefit of the prediction models at different threshold probabilities, the gray solid line represents the net benefit assuming all IgAN (Immunoglobulin A nephropathy) patients develop acute kidney injury (AKI), and the black solid line represents the net benefit assuming no IgAN patients develop AKI.

Lymphocytes, as another component of PNI, have also been shown to be associated with AKI (*Weller, Varrier & Ostermann, 2017*). Lymphocyte count is a marker of cell-mediated immunity, and low lymphocyte counts may be associated with immune suppression and increased inflammatory activity, although the effects of different types of lymphocytes on inflammation and renal function may be contentious (*Kurtul, Gok & Esenboga, 2021*).

The interplay between malnutrition and inflammation is complex, playing a crucial role in the pathogenesis of AKI (*Liu et al., 2023*; *Stumpf et al., 2023*). Thus, we studied other indices related to inflammation. Ultimately, the inflammation-related predictive factors identified included CRP, SIRI, LMR, PLR, and LCR. As a reliable inflammatory biomarker, CRP is strongly associated with the increased risk and severity of AKI in patients with acute myocardial infarction or group A streptococcal bacteremia (*Cosentino et al., 2019*; *Li et al., 2023*; *Vilhonen et al., 2022*). CRP may promote abnormal repair processes in AKI by facilitating G1 cell cycle arrest *via* Smad3-dependent pathways (*Lai et al., 2016*). Notably, evidence linking CRP to AKI specifically in IgAN patients has been sparse. Our study now provides novel evidence demonstrating CRP's predictive value for AKI occurrence in this distinct population.

Besides CRP, inflammatory cells play a significant role during the initiation, proliferation, and recovery phases of AKI (*Weller, Varrier & Ostermann, 2017*). A clinical study involving septic patients in the intensive care unit found that a reduced LCR is an independent

predictive factor for the occurrence of AKI (*Song, Hu & Zhang, 2024*). A lower level of LMR is associated with an increased risk of postoperative AKI in patients with acute DeBakey type I aortic dissection (*Ma et al., 2021*). Similarly, the PLR is recognized as an independent risk factor for sepsis-related or contrast-induced AKI (*Velibey et al., 2017*; *Zhao et al., 2024*). SIRI, a novel inflammatory index based on peripheral neutrophil, monocyte, and lymphocyte counts (*Qi et al., 2016*), has been shown to be independently associated with AKI (*Chen et al., 2024a*). However, few studies have systematically evaluated these inflammatory indices as AKI predictors specifically in IgAN. In our study, we integrated various white blood cell counts, platelet counts, and CRP levels into different indices to construct multiple prediction models for AKI occurrence in the IgAN population.

Inflammation is intricately linked to both IgAN and AKI (*Cao et al., 2024*). IgAN is an inflammatory disease characterized by the deposition of immune complexes composed of galactose-deficient IgA1 (Gd-IgA1) and autoantibodies in the mesangial region, which leads to mesangial cell proliferation and excessive mesangial matrix production (*Luvizotto et al., 2022*). This process initiates a pro-inflammatory cascade: activated mesangial cells secrete IL-6 and TGF-$\beta$, stimulating leukocyte recruitment and ultimately inducing programmed cell death in mesangial cells, podocytes, and renal tubular epithelial cells (*Zhai et al., 2024*). There is also a complex interplay between AKI and inflammation. On the one hand, inflammation plays a significant role in the initiation and progression of AKI (*Fu et al., 2022*). On the other hand, dying tubular cells may trigger a secondary inflammatory response, further amplifying tubular cell death (*Guerrero-Mauvecin et al., 2023*). When the kidneys are subjected to harmful stimuli, immune cells and renal cells recognize the damage and secrete cytokines and chemokines (*Jang & Rabb, 2015*; *McWilliam et al., 2021*). This process can recruit leukocytes from the circulation to the damaged area to combat invading pathogens, repair injured tissue, and restore tissue homeostasis (*Jang & Rabb, 2015*; *McWilliam et al., 2021*). However, excessive inflammatory responses can induce tissue damage, leading to the development of AKI (*McWilliam et al., 2021*). Neutrophils respond rapidly to injury and can interact with endothelial chemokines and adhesion molecules, playing a role in the early stages of renal damage (*Fu et al., 2022*). Subsequently, monocytes migrate to the injured kidneys and differentiate into macrophages. By secreting pro-inflammatory cytokines such as IL-6, tumor necrosis factor-$\alpha$, and IL-1$\beta$, they further infiltrate the tissue and exacerbate renal damage (*Luo et al., 2023*). Macrophages can differentiate into M1 macrophages, which have pro-inflammatory effects, and M2 macrophages, which exert anti-inflammatory actions (*Li et al., 2024*). M2 macrophages are crucial for tissue remodeling and repair; however, if this process is dysregulated, it can lead to renal tissue fibrosis and chronic kidney disease (*Bonavia & Singbartl, 2018*). In addition, there is a close relationship between coagulation and inflammatory responses (*Xiao et al., 2022*). During the occurrence of AKI, platelets can promote inflammation by stimulating endothelial cells and recruiting and activating leukocytes (*Jansen, Florquin & Roelofs, 2018*). Aggregated platelets, along with activated leukocytes and fibrin, promote microthrombus formation in the renal microvasculature, which can obstruct renal blood vessels and glomeruli, leading to renal ischemia and damage (*She et al., 2023*). In the future,

regulating inflammation may prove to be an effective strategy for reducing the risk of AKI in IgAN patients.

Our study identified interstitial fibrosis/tubular atrophy (T) as a clinical predictor for AKI in IgAN. According to the Oxford classification for IgAN, different degrees of interstitial fibrosis/tubular atrophy are scored as T0 (≤25%), T1 (26%–50%), and T2 (>50%) (*Trimarchi et al., 2017*). Existing studies have shown that in IgAN patients, the proportion of tubular atrophy and interstitial fibrosis is higher in the AKI group compared to the non-AKI group (*Zhang et al., 2016*; *Zhang, Zhuang & Liao, 2018*). Our findings are consistent with previous studies. After the onset of AKI, renal tubular cells may stimulate fibroblasts *via* paracrine signaling, leading to interstitial fibrosis and further exacerbating tubular damage in IgAN (*Livingston et al., 2023*). Therefore, when exploring the risk of AKI in IgAN patients, special attention should be paid to the T score in renal biopsy pathology features.

## Differences in variables compared to previous studies

Previous studies have reported a strong association between macroscopic hematuria and the occurrence of AKI in IgAN patients (*Sevillano et al., 2023*). However, this factor was not included in our model. One possible reason for this outcome is that patients with macroscopic hematuria may be identified and treated more promptly in clinical settings. Additionally, our study did not record the severity and duration of macroscopic hematuria.

Moreover, it is worth noting that patients with AKI had lower serum IgG and IgM levels in our study. A previous study found that serum IgG levels in IgAN patients decrease as the fluorescence intensity of IgG in kidney biopsies increases, indicating a negative correlation between serum IgG levels and renal IgG deposition (*Dong et al., 2018*). Another retrospective cohort study found that IgAN patients with elevated serum IgG levels had a higher cumulative renal survival rate (*Liu et al., 2019*), which also indicated that a decrease in serum IgG levels at the time of kidney biopsy is independently associated with poor renal outcomes in IgAN patients (*Liu et al., 2019*). The decrease in serum IgG may result from consumption due to glomerular deposition, urinary loss, and increased catabolism (*Tsai et al., 2019*).

## Strengths and limitations

Our study developed seven prediction models for AKI occurrence in IgAN patients, ranging from a complex model with 11 variables to a simplified model comprising only eGFR, BUN, and CRP. This provides clinicians with options to choose the suitable model based on specific circumstances. Our study has several strengths. Firstly, we proposed multiple prediction models for AKI occurrence in patients with IgAN, providing new insights and reference methods for early prevention of AKI in this population. Secondly, we identified new risk factors and predictors for AKI in IgAN, such as inflammation and nutritional indicators. These indicators are easily assessed readily available, and cost-effective, offering significant clinical value for improving IgAN patient's outcomes.

Yet, our study has limitations. Firstly, being a retrospective study with a relatively small sample size, this research is limited in the capacity of identification, controlling of confounding factors, and conducting in-depth stratified analyses. For example, this study

did not include direct measures of nutritional status, such as dietary salt/protein intake, skeletal muscle mass, or muscle strength. Consequently, the accuracy and reliability of the model may be compromised. Future studies will focus on expanding the sample size and collecting relevant data to further improve the model's reliability and applicability. Secondly, AKI was only diagnosed by creatinine changes in this study, which may lead to an underestimation of AKI incidence. Thirdly, our renal biopsy cohort only included M1 lesions, which may limit the generalizability of the findings to M0 patients. Finally, the model development and validation were based on clinical-pathological data from a single institution. This may introduce geographic or other systematic biases. Future studies will require the use of multi-center data for external validation to enhance the generalizability of the results.

## CONCLUSIONS

CRP, SIRI, LMR, PLR, LCR, and PNI were identified as novel AKI predictors in patients with IgAN, highlighting the critical role of inflammatory and nutritional status. A series of prediction models incorporating these factors were developed for better clinical applicability, with the full model performing the best.

### Funding

This research was funded by the Jilin Tianhua Health Public Welfare Foundation, with the project number Z2022LJL009. The funders had no role in study design, data collection and analysis, decision to publish, or preparation of the manuscript.

### Grant Disclosures

The following grant information was disclosed by the authors:
Jilin Tianhua Health Public Welfare Foundation: Z2022LJL009.

### Competing Interests

The authors declare there are no competing interests.

### Author Contributions

- Huimin Li conceived and designed the experiments, performed the experiments, analyzed the data, prepared figures and/or tables, authored or reviewed drafts of the article, and approved the final draft.
- Chuyue Qian analyzed the data, authored or reviewed drafts of the article, and approved the final draft.
- Jingda Huang analyzed the data, prepared figures and/or tables, authored or reviewed drafts of the article, and approved the final draft.
- Mindan Sun conceived and designed the experiments, authored or reviewed drafts of the article, and approved the final draft.

## Human Ethics

The following information was supplied relating to ethical approvals (i.e., approving body and any reference numbers):

The study involving human participants was approved by the Ethics Committee of the First Hospital of Jilin University (No. 2024-442). This research was conducted in accordance with local legislation and institutional requirements. The ethics committee/institutional review board waived the requirement for written informed consent from participants or their legal guardians/next of kin, as the data analyzed in our retrospective study was completely anonymous. Individuals whose data was being studied had no direct involvement or influence. Obtaining informed consent from each participant was deemed impractical and unnecessary, as the study posed no risks or potential harm to the subjects.

## Data Availability

Raw data is available in the Supplemental Files.

## Supplemental Information

Supplemental information for this article can be found online at http://dx.doi.org/10.7717/peerj.19917#supplemental-information.

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
