# Peer review of "Inflammatory and nutritional indexes as predictors of acute kidney injury in patients with Immunoglobulin A nephropathy: a retrospective study"

_PeerJ, doi:10.7717/peerj.19917_

## Round 0.1 · original submission · Major Revisions

Please address the concerns of both reviewers and amend the manuscript accordingly.

Reviewer 1 ·

Basic reporting

-

Experimental design

-

Validity of the findings

-

Additional comments

This study presents a risk prediction model for AKI in patients with IgA nephropathy, incorporating indices related to inflammation and nutritional status. The clinical applicability of such a model is promising. However, many of the indices used—such as the systemic inflammatory response index (SIRI) and the prognostic nutritional index (PNI)—are composite scores derived from multiple parameters. As such, they may introduce mathematical confounding and obscure the individual contribution of each component.

To establish the independent utility of these indices, it is essential to first evaluate the predictive performance of each constituent variable separately. Demonstrating that the composite index offers superior predictive value compared to its individual components would strengthen the rationale for its inclusion.

Furthermore, the analysis would benefit from incorporating absolute markers of inflammation (e.g., serum IgA levels, complement C3) and more direct measures of nutritional status, such as dietary salt and protein intake, skeletal muscle mass, or muscle strength. Including such variables may enhance the clinical validity and interpretability of the model.

Reviewer 2 ·

Basic reporting

-

Experimental design

-

Validity of the findings

-

Additional comments

The article was written correctly, and my comments are primarily technical.
The introduction properly introduces the reader to the issues of the article.
The purpose should be better formulated.
I have no reservations about the research. Here, everything is consistent and properly defined. I appreciate the "limitations of the study".
I have a question, was AKI pre-/intra/post-renal? I am unsure if I understood the researchers' idea exactly: they were trying to predict the occurrence of AkI in the group of patients with IgAN (?)

In what period were the patients observed before the occurrence of AKI? Are these results presented in Table 1 from the day of admission to the hospital?

The article contains much information about the methodology, but I still do not know how the research group was recruited and what happened next. Besides, I kindly ask the researchers to insert a figure with information on how they created the research group step by step and how it was studied.

I have reservations about the discussion. It is written in one go and is very difficult to get through.
The researchers should divide the discussion into subsections. They should also shorten it.

It would also be helpful to describe these seven models in detail, assess their usefulness, and compare them.
All of this is presented in the article, but it isn't easy to get through it.

The conclusions are a summary, not actual conclusions - I suggest correcting this a bit.

I want to draw attention to stylistic and grammatical errors. I ask the authors to re-read the article with this in mind.

For example: "The present study was conducted to create such tools containing inflammatory and nutritional indexes" - instead of "creating" it should be "create".

Finally, I suggest that the authors add a figure, or a graphical abstract, summarizing the most important discoveries. This will allow the reader to better understand the subject matter presented by the researchers.

---

## Round 0.2 · accepted · Accept

All issues pointed out by the reviewers were adequately addressed, and revised manuscript is acceptable now.